# Influence of Machining Conditions on Micro-Geometric Accuracy Elements of Complex Helical Surfaces Generated by Thread Whirling

**DOI:** 10.3390/mi13091520

**Published:** 2022-09-14

**Authors:** Vasile Merticaru, Gheorghe Nagîț, Oana Dodun, Eugen Merticaru, Marius Ionuț Rîpanu, Andrei Marius Mihalache, Laurențiu Slătineanu

**Affiliations:** 1Department of Machine Manufacturing Technology, Gheorghe Asachi Technical University of Iasi, 700050 Iasi, Romania; 2Department of Mechanical Engineering, Mechatronics, and Robotics, Gheorghe Asachi Technical University of Iasi, 700050 Iasi, Romania

**Keywords:** helical surfaces, thread whirling, thread accuracy, influence factors, numerical modeling and simulation, experimental research, empirical mathematical models

## Abstract

Complex surfaces such as helical ones are commonly used in machinery. Such surfaces can be obtained by various machining processes, one of these processes being thread whirling. The influence of machining conditions needs to be better understood to develop a more precise prediction of the specific resulting errors involved in thread whirling. This paper firstly presents the theoretical conditions which generate micro-deviations on whirled surfaces. A theoretical model which considers the geometrical parameters describing the whirling head and cutters and the process’s whole kinematics was developed. The threaded surface was described as a complex compound surface resulting from intersecting successive ruled helical surfaces corresponding to the cutting edges of the set of cutters from the whirling head. Numerical simulation results were exemplified and validation experiments were both designed and performed. Empirical mathematical models were established to highlight the influence of the input factors such as thread pitch and external diameter, the ratio between the diameter of cutters’ top edge disposal and the thread’s external diameter, the rotary speed of the whirling head, and the rotary speed of the workpiece on some accuracy elements and roughness parameters of the threaded surface.

## 1. Introduction

Helical surfaces represent one of the major categories of complex surfaces commonly used in machinery and generally used in industrial products for materializing functional or aesthetic shapes. Different technological solutions have been developed and are well known for obtaining such surfaces which depend on a large set of characteristics of the particular product and surface to be processed. Mastering the industrial process variables involved (in the sense of identification, inventory, classification, and quantification of influence factors in direct relation to process performance parameters and further optimal adjusting of processing conditions) is vital for final product quality and market value. Three major subjects are considered to be important and are briefly reviewed below for placing the research study in this broader context:Complex surfaces in mathematics and engineering;The current state of the research field with initial discussions on referential approaches and eventually on controversial and diverging hypotheses.The current problems, purpose, and objectives of this research study.

### 1.1. Complex and Helical Surfaces in Mathematics and Engineering

The term “complex surface” can be discussed depending on the context, but its mathematical definition should be firstly considered. In mathematics, a surface is considered a compact, connected two-dimensional manifold where the last represents a set of points or topological space that is modeled on Euclidean space. The Enriques-Kodaira classification divided complex surfaces into ten classes [1]. Within complex surfaces, Krivoshapko and Ivanov mathematically described and classified a great number of helical surfaces, considered them as being generated by a generatrix curve in its motion along a helical directrix and grouping them into two major classes (“ordinary” and “of variable pitch” helical surfaces) [2].

Within both mathematical and engineering contexts, the term “complex surface” has to be discussed in the context of the representation methodology in geometric modeling and CAGD (computer-aided geometric modeling), where surfaces can be classified after the method by which they are generated as geometric entities (as Horvath and Rudas proposed [3]).

Considering the three most well-known techniques for geometric modeling [4] by primitives and Boolean operations, models specific to CSG (constructive solid geometry), and also by boundary representation (B-Rep models), compound surfaces resulting from applying Boolean operations to elementary surface entities can be further included within complex surfaces and are widely used in engineering and technology.

On the other hand, again, from an engineering perspective, complex surfaces in general and helical surfaces in particular which are used to materialize functional or aesthetic shapes for industrial products or machinery should be discussed in terms of utility and technological appropriateness, affordability, and sustainability.

Somarriba-Sokolova et al. discussed the application of some types of ruled helical surfaces in engineering design [5], analyzing and comparing right and oblique helicoids, developable helicoids, pseudodevelopable general type helicoids, and pseudodevelopable helicoids in terms of importance for CAD/CAE/CAM design and their advantages in manufacturing (such as manufacturing easiness, efficiency in material usage, and cost efficiency). The discussed examples were mainly from architectural design.

More specific to machinery and technology, Kostyuk and Barbelko divided complex surface applications into two classes: actual parts with such complex shapes and tools for reproducing similar parts. Undertaking the classification of helical surfaces after Lukshin, they considered that complex helical surfaces are those characterized by variable generatrix and variable pitch and also discussed the issue of helical surfaces in the design of cutting tools, focusing on spherical mills (which are widely used on CNC machine tools) and on grinding wheels (which are mainly used for machining helical surfaces) [6]. Compound helical surfaces were not considered in their study.

Modern technology practically allows for obtaining various types of ordinary or complex helical surfaces on machinery parts such as leading shafts, extrusion screws, worms, screw rotors, etc., or even bone screws and other medical implants. Such technologies start with casting and die forging, cover machining operations such as turning, milling, grinding, rolling, and thread whirling, go further with injection molding or blow molding and finish with additive manufacturing solutions.

### 1.2. Referential Approaches

Within the above-discussed broad context, the scope of the research study covers the area of complex helical surfaces generation by thread whirling.

Thread whirling, also named by others whirlwind milling, is widely recognized as a valuable solution for machining difficult threads. The principle of thread whirling technology supposes a tool holder ring having mounted cutters on its inner side, in radial position, which rotates at high speed around an axis inclined with the lead angle and eccentrically to the axis of the workpiece, the last being in slow feed rotation. The tool head also has a longitudinal feed movement correlated to workpiece rotation, generating the required lead. The thread whirling process could be realized on special dedicated machine tools or standard or CNC lathes endowed with special equipment.

The technological particularities of thread whirling, such as discontinuous chip removal, with longer and thinner chips than in standard milling, less cutting forces, high metal removal rate, and generally dry machining result in a set of benefits such as low machining time, better surface quality, increased thread accuracy, less energy consumption, and cleaner production, which make thread whirling an ideal single and final process for generating long, complex and high-helical threads (particularly in difficult to machine materials and for a large range of parts such as gear worms, steering worms, rack, and pinion spindles, eccentric screws and worms, ball screws, pump screws, pump rotors, extrusion worms, single and multi-lobe motor rotors, medical implant screws, etc.).

On the other hand, the complexity of the thread whirling process could be considered as bringing difficulties in mastering the machining performance. A large set of characteristics, such as the elements of thread geometry transposed in tools geometry, process kinematics elements, mechanical properties of the workpiece material, and positioning setting parameters of the whirling head, can be considered input factors in the thread whirling process.

In the last half of a century, since Burgsmuller GmbH introduced and patented whirling technology [7], many types of research have been developed regarding various aspects of this complex machining process, which one may group as follows:Development of thread whirling equipment and tooling;Applicability and advantages of thread whirling for various complex helical surfaces generation;Inventory and analysis of specific process parameters;Studies on the chip formation and the process mechanics;Theoretical modeling and numerical simulation for specific aspects involved in the thread whirling process;Modeling and simulation for thread whirling tooling;Investigations on accuracy and surface quality, resulted in processed parts to the influence parameters;Research the thermal and energetic phenomena in the whirling process for sustainable development.

Leistritz GmbH is another experienced producer of thread whirling machines [8], providing performance characteristics for high pitch threads, with helix angles up to ±50°, temperature-controlled whirling unit for thermal stability, and improved thread profile accuracy in conditions of dry and wet machining of hard and soft materials.

Concerning the thread whirling technology for bone screws and other medical implant production, GenSwiss [9] is a very important developer and provider of whirling heads, whirling rings, whirling inserts, and, more important, reliable whirling technological data. Their studies revealed valuable inventory and quantification of thread whirling process parameters and comparative discussions on the advantages of thread whirling to other threading processes such as single-point threading, grinding, or rolling. Cheng et al. tested a new method of applying a cutting fluid jet in the internal threading at one end of the dental implants by experiments on the Swiss-type lathe, Star SR-20R [10].

Serizawa et al. developed a micro whirling machine on which 30 μm microgrooves have been machined on 0.3 mm diameter stainless steel wires [11].

Giacomozzi and Turci developed a large and complex applied research on the whirling process in the production of worm gear drives [12]. Their study presented in detail the particularities of the machining process, together with an analysis of chip formation, and a comprehensive comparison, based on experimental data, of time, cost accuracy, and surface quality benefits between thread whirling as technology changes and previous industrial practice on standard machine tools. They also discussed geometric constraints involved in worm shaft design due to the whirling process and, finally, process simulation as motion analysis using SolidWorks multi-body 3D CAD system was proposed. Their research data highlighted whirling a better surface quality with roughness parameter *R_a_* values of 0.5–0.6 μm and even up to 0.4 μm, better accuracy of the workpiece by at least 2 points DIN 3974 or ISO 3408-3, a lower working time, more than 300% increase of productivity, a reducing of the required energy by 23%, and better environmental sustainability due to the removal of cutting oil.

Laprade [13] particularly discussed the applicability and advantages of thread whirling on CNC Swiss-type lathes and the advancements in thread whirling tooling, concluding that although live tooling drives can provide speeds from 5000 to 10,000 rpm, thread whirling tends to require only 2000 to 3000 rpm. Also, the advantages of some solutions of coolant-through whirling attachments and cutter rings were discussed.

The areas of application for complex helical surfaces on parts from various industries, the advantages of thread whirling, and the types of threads that can be obtained, together with some performance data in terms of productivity in comparison with single point threading, were reviewed on the internet platform Interempresas-Metalworking [14].

Particularly for medical implant production, Rakowski showed that every bone screw manufacturer develops a custom thread type for its screws and that, in such a situation, thread whirling is highlighted as the best alternative to single-point threading [15].

Soshi et al. proposed a directly-driven thread whirling unit with advanced tool materials to mass-produce implantable medical parts [16]. They validated the solution by experimenting with a thread whirling of titanium bone screws on a DMG MORI SPRINT 20|8 linear automatic lathe with a Swiss-type kit, compared with a conventional mechanical whirling unit. The new solution has proved higher cutting speeds, minimized vibration, and no backlash.

A study on the inventory and classification of the process variables for worm screw production using thread whirling devices was developed by Cretu [17] based on systemic analysis principles. Uncontrollable process variables, representing perturbation factors that can determine noise affecting in experiment results, were not discussed. A theoretical approach for determination of the theoretical deviations at the tooth bottom processing with a thread whirling device was also proposed by Cretu [18], adapting a much older mathematical model proposed by Snaiderman at the half of the previous century and customizing it for worm screw generation. The dependencies of these deviations on the worm constructive parameters, the threading device’s constructive characteristics, and the working regime were studied using dedicated software. The simplifying hypothesis of not considering the axial feed in the mathematical model affects the significance of the theoretical results.

Diachun et al. [19] developed a similar approach to theoretical modeling for investigating the geometrical parameters in whirling screw surfaces. Although the model considered the movement of the whirling head along the workpiece, its accuracy and signification were affected by the simplifying hypothesis of considering a single generation point from the cutting-edge profile (of constant radial position on the tool holder ring).

Song et al. proposed a theoretical model for the whirling process based on equivalent cutting volume [20]. The cutting force and the chip morphology were investigated to validate the model. The cutting force simulation resulted in good agreement with the experimental results, with errors being less than 16.5%. The research also studied the effect of the tool edge geometry on cutting forces. However, as the authors admit, the model is simplified, based on the simplifying hypothesis that the trajectory is straightened, and the tool reference plane is always perpendicular to the flat bottom. A chip with saw-toothed edges was obtained from simulation to approximate the real comma-shaped chips resulting in whirling.

Mohan and Shunmugam developed a mathematical model for simulating the whirling process and tool profiling for worm machining [21]. Modeling was conducted using 3D coordinate geometry and applying homogenous coordinate transformations of discretized surface coordinate points. The case studies referred to two types of worms, named by the authors the straight-sided in axial section and the involute helicoid, respectively. The effect of tool head tilting was also analyzed. The complexity of the modeling and simulation approach for whirling can be compared with a much simpler simulation study developed by Albu [22] for a double-start worm processed by milling with a cylindrical-frontal tool.

Another research study focused on mathematical modeling and numerical simulation of the thread whirling process with standard cutters was carried out by Han, and Liu [23], where the complex helical surface of a screw shaft was mathematically modeled based on its axial section profile, defined through several sampled points, and the errors along the axial section profile and along the cross-section profile were assessed. The model was tested on a case study referring to a double-start screw with symmetrical helical grooves.

A reasonable theoretical dynamic model for stability prediction in blade whirling was proposed by Han et al. [24]. The model fully considered the geometric immersion of the whirling cutter and the dynamic parameters of the workpiece system. They showed that the material removal changes the natural frequency and modal shape of the system, which causes the variation of the limit on cutting depth.

Zanger et al. proposed and compared numerical and analytical methods for modeling cutting tool profiles for conventional and synchronized whirling [25]. Their numerical method used the calculation of intersections of a workpiece surface and a cutting tool blank. The proposed model was model less flexible when adopted for different workpieces.

Yi et al. approached the problem of modeling and simulation for thread whirling tooling [26]. They also proposed a machining strategy for complex helical surfaces of rotors and stators of positive displacement motors, based on combining whirling with the capabilities of a 5-axis CNC rotary-table horizontal milling machine. The model idea was to adjust the rotational symmetric surface of the milling cutter tip and the principal curvatures of the milled surface to create a close fit. The particularities of the whirling generation method can be highlighted if comparing the study of Yi et al. to that developed by Kuang et al. [27] for modeling and simulation of helical surfaces of screw rotors generated by milling with disk mill cutters, the modeling strategy basing on a minimal orientation-distance algorithm with the spatial discretization method.

Particular experimental investigations on accuracy and surface quality resulting in whirling processed parts to the influence parameters have been published by several researchers. Among them, Cretu and Negoescu [28] performed experimental tests by whirling worm shafts on the non-alloy quality structural steel E335-EN 10025- 2:2019, with a hardness of 224 HB, the dependence of the roughness parameter *R_a_* on the worm modulus, on the tool speed, and the feed per tooth, at single pass machining in the feed direction, was studied. Values in the range *R_a_* = 0.25–1.2 μm were obtained, which is comparable to grinding operation.

Similarly conducted experiments by Paraschiv et al. [29] for trapezoidal thread driving shafts, comparatively on non-alloy steel for quenching and tempering C45 (SR EN ISO 683-1:2018) and on structural steel S460JR (SR EN 10025-2:2019), studying obtained roughness dependence on the machining conditions, proved again that thread whirling could be successfully applied as a single operation for processing such kind of helical surfaces.

A much more complex research approach was developed by Guo et al. [30,31,32,33,34]. They comparatively studied the surface roughness and the microstructure deformation for whirling on GCr15 material, modeling surface roughness for predicting the effects of cutting parameters on it and experimenting for model validation, and investigating microstructure evolution using electron backscatter diffraction (EBSD) [30]. Experiments on the residual stress resulted when whirling large screws were further conducted [31]. They highlighted the cutting depth as the dominant factor, and the interaction between cutting speed and tool number was found significant. Another set of their experiments studied three characteristic parameters of residual stress: residual surface stress, maximum residual stress, and residual stress extreme value, together with the deformation behavior for whirling large screws [32]. Surface roughness and tangential cutting force were also studied [33] to achieve process multi-optimization. The cutting vibration was modeled, and the deflections were further considered in surface topography modeling [34].

Son et al. also studied the cutting forces of the whirling process [35]. They used a model of un-deformed chip thickness and the DEFORM software to estimate the values of the cutting forces, and the effects of cutting forces on the tool were analyzed using the ADAMS software. Son et al. also analyzed the whirling unit’s temperature distribution, thermal deformation, and thermal stress [36].

Research on the thermal and energetic phenomena for whirling processes was also performed in directions such as the simulation of the thermal elongation error of whirlwind hard milling ball-screws, as carried out by Li et al. [37]; modeling the material removal volume and cutting forces for predicting specific cutting energy, respectively, validating the theoretical models by forces and power measurements, as carried out by He et al. [38]; research on temperature distribution in lead-screw whirling milling considering the transient un-deformed chip geometry, also developed by He et al. [39]; and investigations on workpiece temperature and surface topography, as carried out by Liu et al. [40].

Researchers’ results generally highlighted whirling as a less energy consumption process than other threading solutions and as a cleaner production option, in the dry machining variant.

Another complex research approach on ball screw shafts whirling belongs to Wang et al. [41,42]. They investigated the material removal mechanism in the whirling process, thus predicting process variables such as the un-deformed chip geometry, material removal rate, and cutting forces [41]. The authors found the proposed mathematical model aprecise precise, considering geometry, kinematics, and mechanics, and also provided a prediction of circularity error, scallop height, and surface roughness as a function of tools and workpiece motion, position, and dimension parameters. The validation experiments proved the largest error, 11.3%, and an average error of 13.0% for cutting force and surface roughness prediction. They also modeled and analyzed the specific cutting energy in the whirling process [42].

The dynamics of the whirling process were investigated by Wang et al. by dynamic modeling with Deform 3D software [43] and optimization of dynamic performances of large-scale lead screws whirling [44]. A measurement system based on the use of laser for the evaluation of the profile accuracy in the case of a double-headed screw rotor was analyzed by Dong et al. [45].

### 1.3. Current Problem: Purpose and Objectives of the Research

From analyzing the above considerations on the current state in the research field, the importance of machining accuracy for complex helical surfaces generated by thread whirling came out as a covering conclusion due to their essential functional requirements.

Most previous research approaches focused mainly on helical surfaced parts such as worm shafts, ball screws, motor rotors, or medical implant screws.

Another category of threads used on screws in some leading mechanisms construction, such as in machine tools driving systems or in some industrial valves, is represented by the trapezoidal threads. Therefore, the development of research studies upon the influence of machining conditions on micro-geometric accuracy elements for trapezoidal threads machined by thread whirling was considered a significant current problem to be approached.

More than that, previous particularly customized approaches in theoretically modeling the complex helical surfaces generated by whirling, some of them based on simplifying hypotheses that affect the models’ accuracy, highlighted the need for a more deep theoretical approach in the development of a reliable modeling algorithm, which must consider both the complex mathematical definition and the principles of representation in geometric modeling for such surfaces, that actually can be described as compound surfaces resulted from intersecting successive ruled helical surfaces corresponding to the cutting edges of the set of cutters from the whirling head.

Under the purpose of studying the influence of machining conditions on micro-geometric accuracy elements of complex helical surfaces generated by thread whirling, the objectives of the particular research approach focused on two directions:Theoretically modeling and numerically simulating the whirled complex helical surfaces for reasonable prediction of specific machining errors, based on less simplifying hypotheses and taking into account all of the geometrical parameters describing the whirling head and cutters and also the whole process kinematics;Model validation through experiments and finding empirical mathematical functions to describe the influence of machining conditions on some accuracy elements for whirled trapezoidal threads.

## 2. Research Study—Materials and Methods

### 2.1. Theoretical Conditions in Surface Generation at Thread Whirling

The research study started with a systematic identification, inventory, and classification of the whirling process variables. Thus, the real machining process was analyzed by delimitating a set of characteristic zones for process variables’ inventory and classification, as shown in Figure 1.

Here, *zSP* is the zone of the whole technological system, *zMU-DFV* is the zone of whirling equipment, *zSA* is the cutter zone, *zSmf* is the zone of the blank, *zP* is the zone of the threaded part, *zA* is the chip removal zone, *zMA* is the zone of the eventual cooling system, *n_c_* is the rotary speed of whirling head (rpm), *n_p_* is the rotary speed of workpiece (rpm), *f_L_* is the longitudinal feed rate, depending on thread pitch (mm/rev), and *p* and *Q* are the pressure and flow rate parameters of eventual cooling agent, respectively.

Theoretical conditions in surface generation at thread whirling have been researched under the following hypotheses:The threaded surface generated by whirling represents a complex compound surface resulting from intersecting successive ruled helical surfaces corresponding to the cutting edges of the set of cutters from the whirling head;The geometrical parameters describing the whirling head and cutters, such as the thread diameters, the number of cutters on the whirling head, the tool edge profile geometry, the diameter or the radius of cutters’ top edge disposal on the whirling tool holder, the eccentricity of whirling head axis, the whirling head tilting angle and also the whole kinematics of the process, respectively, the rotary speed of whirling head, the rotary speed of workpiece, and the axial feed rate were considered as influential factors in modeling;The process dynamics are not considered in the modeling algorithm, nor are the thermal or elastic deforming phenomena.The modeling approach involved the following steps:Idealized helical surface (*Σ*_0_), theoretical helical surface (*Σ_q_*), and intersection curve (*Cq*), between two successive theoretical surfaces (*Σ_q_*) and (*Σ*_*q*+1_) were mathematically represented [46] based on the geometrical schemes (as in Figure 2).

The symbols from Figure 2 represent as follows: *p* is the thread pitch (mm), *d* is the thread external diameter (mm), *d*_2_ is the thread pitch diameter (mm), *d*_3_ is the thread internal diameter (mm), *α* is the thread profile’s angle (°), *v_l_* is the axial feed speed (mm/min), (*Г*_0_) is the thread generatrix, (Δ_0_) is the thread helical directrix, *φ_p_* is the rotation angle for the part in coordinate transformation (°), *ω_p_* is the angular speed of part rotation (s^−1^), (*Г*) is the cutter edge generatrix, (Δ) is the whirling helical directrix, *R* is the radius of cutters’ top edge disposal (mm), *β* is the whirling head tilting angle (°), *φ_c_* is the rotation angle for the whirling head in coordinate transformation (°), *e* is the eccentricity of whirling head axis (mm), *Ψ_c_* is the angular step of cutters’ disposal (°), *Ψ* is the angular period of theoretical surface generation (°).

Figure 2a shows the standard generator profile for trapezoidal threads. Here, *O*_0_*x*_0_*y*_0_*z*_0_ is the local (mobile) coordinate system, and *Oxyz* is the general (fixed) coordinate system on the thread axis. The mathematical description of the generator profile is based on representation from Figure 2b. Generation of idealized helical surface (*Σ*_0_) was considered as in Figure 2c, by sweeping the cutter edge generatrix (*Г*) along the whirling helical directrix (Δ). For (*Σ*_0_) modeling, three coordinate transformations were applied, as in Figure 2d, respectively: origin translation from *O*_0_ to *O*’, rotation of *φ**_p_* angle around *O*’*x*’ axis, and origin translation from *O’* to *O*, where *O*’*x*’*y*’*z*’ is an intermediate coordinate system.

For the flanks, which are considered of major importance for the trapezoidal threads as leading mechanisms, the idealized helical surface (*Σ*_0_) was mathematically described as follows:
(1)$$(\Sigma_{0}):\;\left\{ \begin{array}{l} x=\pm \frac{p}{4}\pm \left(y_{0}-\frac{d_{2}-d_{3}}{2}\right)\cdot tg\frac{\alpha }{2}+\frac{p\cdot n_{p}\cdot t}{60}\\ y=(y_{0}+\frac{d_{3}}{2})\cdot \cos(\omega_{p}\cdot t)\\ z=(y_{0}+\frac{d_{3}}{2})\cdot \sin(\omega_{p}\cdot t) \end{array}\right.$$

For (*Σ_q_*) modeling, six-coordinate transformations were applied, as in Figure 2e, respectively: origin translation from *O_0_* to the center of cutters’ disposal *O_c_*, rotation of *φ_c_* angle around the axis of the whirling head, rotation of *β* angle around *O_c_y*_1_ axis corresponding to whirling head tilting, origin translation from *O_c_* to *O_4_* corresponding to longitudinal feed, origin translation from *O_4_* to the workpiece axis in *O_p_*, and rotation of *φ_p_* angle around the workpiece axis.

Again, for the flanks, the theoretical helical surface (*Σ_q_*) generated by a cutter of *q* order, for the technological variant of whirling against the circular feed, was given by:(2)$$(\Sigma_{q}):\;\left\{ \begin{array}{l} x=a_{3}\cdot \left(\pm a_{4}\pm y_{0q}\cdot a_{1}\right)+a_{0}\cdot (y_{0q}+R)\cdot \sin\left(\omega_{c}\cdot {t_{q}}^{\prime }\right)+\frac{p\cdot n_{p}\cdot t_{q}}{60}\\ y=a_{0}\cdot \left(\pm a_{4}\pm y_{0q}\cdot a_{1}\right)\cdot \sin\left(\omega_{p}\cdot t_{q}\right)-e\cdot \cos\left(\omega_{p}\cdot t_{q}\right)+(y_{0q}+R)\cdot \\ \left[\cos\left(\omega_{c}\cdot {t_{q}}^{\prime }\right)\cdot \cos\left(\omega_{p}\cdot t_{q}\right)-a_{3}\cdot \sin\left(\omega_{c}\cdot {t_{q}}^{\prime }\right)\cdot \sin\left(\omega_{p}\cdot t_{q}\right)\right]\\ z=-a_{0}\cdot \left(\pm a_{4}\pm y_{0q}\cdot a_{1}\right)\cdot \cos\left(\omega_{p}\cdot t_{q}\right)-e\cdot \sin\left(\omega_{p}\cdot t_{q}\right)++(y_{0q}+R)\cdot \\ \left[\cos\left(\omega_{c}\cdot {t_{q}}^{\prime }\right)\cdot \sin\left(\omega_{p}\cdot t_{q}\right)+a_{3}\cdot \sin\left(\omega_{c}\cdot {t_{q}}^{\prime }\right)\cdot \cos\left(\omega_{p}\cdot t_{q}\right)\right] \end{array}\right.$$
where:(3)$$\begin{array}{c} a_{0}=\sin\beta \\ a_{1}=tg\frac{\alpha }{2}\\ a_{2}=a_{0}\cdot a_{1}\\ a_{3}=\cos\beta \\ a_{4}=\frac{p}{4}-\frac{d_{2}-d_{3}}{2}\cdot a_{1} \end{array}$$

The complex helical surface obtained at thread whirling was defined by intersecting successive ruled helical surfaces corresponding to the cutting edges of the set of cutters from the whirling head. The angular step of cutters’ disposal *Ψ_c_*, as in Figure 2f, was considered for calculating the angular period *Ψ* of theoretical surface generation, as shown in Figure 2g.

Intersection curve (*Cq*), between two successive theoretical surfaces (*Σ_q_*) and (*Σ*_*q*+1_), as in Figure 2h, was generally given by:(4)$$(C_{q}):\;\left\{ \begin{array}{l} a_{3}\cdot x_{0q}+a_{0}\cdot (y_{0q}+R)\cdot \sin\left(\omega_{c}\cdot {t_{q}}^{\prime }\right)+\frac{p\cdot n_{p}\cdot t_{q}}{60}=\\ =a_{3}\cdot x_{0q+1}+a_{0}\cdot (y_{0q+1}+R)\cdot \sin\left(\omega_{c}\cdot {t_{q+1}}^{\prime }\right)+\frac{p\cdot n_{p}\cdot t_{q+1}}{60}\\ a_{0}\cdot x_{0q}\cdot \sin\left(\omega_{p}\cdot t_{q}\right)-e\cdot \cos\left(\omega_{p}\cdot t_{q}\right)+(y_{0q}+R)\cdot \\ \left[\cos\left(\omega_{c}\cdot {t_{q}}^{\prime }\right)\cdot \cos\left(\omega_{p}\cdot t_{q}\right)-a_{3}\cdot \sin\left(\omega_{c}\cdot {t_{q}}^{\prime }\right)\cdot \sin\left(\omega_{p}\cdot t_{q}\right)\right]=\\ =a_{0}\cdot x_{0q+1}\cdot \sin\left(\omega_{p}\cdot t_{q+1}\right)-e\cdot \cos\left(\omega_{p}\cdot t_{q+1}\right)+(y_{0q+1}+R)\cdot \\ \left[\cos\left(\omega_{c}\cdot {t_{q+1}}^{\prime }\right)\cdot \cos\left(\omega_{p}\cdot t_{q+1}\right)-a_{3}\cdot \sin\left(\omega_{c}\cdot {t_{q+1}}^{\prime }\right)\cdot \sin\left(\omega_{p}\cdot t_{q+1}\right)\right]\\ -a_{0}\cdot x_{0q}\cdot \cos\left(\omega_{p}\cdot t_{q}\right)-e\cdot \sin\left(\omega_{p}\cdot t_{q}\right)+(y_{0q}+R)\cdot \\ \left[\cos\left(\omega_{c}\cdot {t_{q}}^{\prime }\right)\cdot \sin\left(\omega_{p}\cdot t_{q}\right)+a_{3}\cdot \sin\left(\omega_{c}\cdot {t_{q}}^{\prime }\right)\cdot \cos\left(\omega_{p}\cdot t_{q}\right)\right]=\\ =-a_{0}\cdot x_{0q+1}\cdot \cos\left(\omega_{p}\cdot t_{q+1}\right)-e\cdot \sin\left(\omega_{p}\cdot t_{q+1}\right)+(y_{0q+1}+R)\cdot \\ \left[\cos\left(\omega_{c}\cdot {t_{q+1}}^{\prime }\right)\cdot \sin\left(\omega_{p}\cdot t_{q+1}\right)+a_{3}\cdot \sin\left(\omega_{c}\cdot {t_{q+1}}^{\prime }\right)\cdot \cos\left(\omega_{p}\cdot t_{q+1}\right)\right] \end{array}\right.$$

The thread flank theoretical generated profile in the axial section plane was mathematically represented by calculating a discretized set of points’ coordinates. Further on, some of the significant generation errors have been modeled.

Within this modeling step, complex systems of transcendent equations involved by mathematical models for the studied error parameters have been numerically solved using the Newton-Raphson method, needing to define approximated solutions. As part of the complex modeling approach, some particular mathematical description aspects have been solved based on geometrical schemes, as in Figure 3.

In Figure 3, the notations represent as follows: *R_Ci_* is the radius of the approximated trajectory of a generator point *M_i_* from the tool cutting edge, *φ_p_*_max*i*_ is the contact half-angle for a generator point *M_i_*, *φ_p_*_0_ is the angular position of the axial section plane considered for the study, *r_min_* is the minimum radius of the theoretical profile generated by a cutter in an axial section plane, *n* is the number of theoretical helical surfaces intersecting a particular axial plane, *φ_pM_* and *φ_pm_* are the extreme angular positions of the first order generative intersection curve (*C_1_*). The other notations are nominated as above.

Concerning Figure 3, the solved modeling aspects were: calculus of the radius *R_Ci_*, approximated and exact calculus of the contact half-angle *φ_p_*_max*i*_; approximated and exact calculus *r_min_* for thread flank and thread groove bottom, calculus of the number of theoretical helical surfaces intersecting a particular axial plane; calculus of the number of intersection points between (*Cq*) curves and the axial plane, on the thread flank, calculus of the exact coordinates of the generated set of points on the thread flank profile and calculus of the studied errors.

The radius *R_Ci_* of the approximated trajectory of a generator point *M_i_*(*x_0i_*, *y_0i_*) from the tool cutting edge was calculated (see Figure 3a) with the formula:(5)$$R_{ci}=\sqrt{{\left(R_{c}+y_{0i}\right)}^{2}+{\left(x_{0i}\cdot \sin\beta \right)}^{2}}$$
where *R_c_* is the radius of cutters’ top edge disposal (mm).

The approximation of contact half-angle *φ_p_*_max*i*_ for *M_i_* (see Figure 3b) was given by:(6)$$\phi_{p\max i}=\arccos\frac{R_{ci}^{2}-\left(r^{2}+e^{2}\right)}{2\cdot r\cdot e}$$
where *r = d/*2 (mm). The exact value for *φ_p_*_max*i*_ was determined by intersecting the helical theoretical curve (*E_i_*) corresponding to the trajectory of the considered generator point with the cylinder of *r* radius. It is supposed to solve the following system of equations:(7)$$\left\{ \begin{array}{l} y(t)=r\cdot \cos\phi_{p\max i}\\ z(t)=r\cdot \sin\phi_{p\max i} \end{array}\right.$$

The approximated value of the minimum radius of the theoretical profile generated by a cutter in an axial section plane *r_min_* (see Figure 3c) was given, for the thread flank, by:(8)$$\begin{array}{c} r_{\min}=R_{ci}\cdot \frac{\sin(\phi_{p0}-\delta )}{\sin\phi_{p0}}\\ \delta =\arcsin\frac{e\cdot \sin\phi_{p0}}{R_{ci}} \end{array}$$

Something similar was carried out for the thread groove bottom (see Figure 3d). The exact value for *r_min_* was determined by intersecting the helical theoretical curve (*E_i_*) corresponding to the trajectory of the considered generator point with the considered axial plane. It was intended to solve the following system of equations:(9)$$\left\{ \begin{array}{l} y(t)=r_{\min}\cdot \cos\phi_{p0}\\ z(t)=r_{\min}\cdot \sin\phi_{p0} \end{array}\right.$$

The calculus of the number of theoretical helical surfaces intersecting a particular axial plane (see Figure 3e) was carried out with the formula:(10)$$n=1+\left[\frac{\phi_{p\max}+\phi_{p0}}{\psi }\right]$$

The number of intersection points between (*Cq*) curves and the axial study plane on the thread flank (see Figure 3f) was given by
(11)$$\begin{array}{c} n=\left[\frac{\phi_{pM}-\phi_{pm}}{\psi }\right],\;\mathrm{for}\;\phi_{p0}-\phi_{pM}<\psi -\left(\phi_{pM}-\phi_{pm}\right)+\left[\frac{\phi_{pM}-\phi_{pm}}{\psi }\right]\cdot \psi \\ n=\left[\frac{\phi_{pM}-\phi_{pm}}{\psi }\right]+1,\;\mathrm{for}\;\phi_{p0}-\phi_{pM}\geq \psi -\left(\phi_{pM}-\phi_{pm}\right)+\left[\frac{\phi_{pM}-\phi_{pm}}{\psi }\right]\cdot \psi \end{array}$$
where *φ_pM_*, *φ_pm_* define the angular positions of the extremities of the curve (*C_1_*).

The numerical calculus and simulation on the developed models have been performed with a customized engineering software tool [47], developed by modular, procedural programming under MATLAB.

Theoretically modeling and numerically simulating the complex helical surfaces generation at thread whirling, as presented above, allowed the study of several micro-geometric accuracy elements such as the following: the profile errors on the thread flank, in axial section plane, symbolized as *E_pax_* (mm); the profile error on the pitch diameter, symbolized as *E_pdm_* (mm); the scallop height on the thread flank, in axial section plane, symbolized as *h_max_* (μm); and the local error of the helix pitch, symbolized as *E_ph0_* (mm). As influence factors upon these theoretical errors, the following set of machining conditions were considered: the thread pitch *p* (mm), the thread external diameter *d* (mm), the number of cutters on the whirling head *z_c_*, the diameter coefficient of whirling process *k_d_* defined as the ratio between the diameter *D* of cutters’ top edge disposal, and *d*, the whirling head tilting angle *β* (°), the rotary speed of whirling head *n_c_* (rpm) and the rotary speed of workpiece *n_p_* (rpm). Hereby, the distribution of the profile errors *E_pax_* along the thread flank highlighting the influence of parameters *z_c_* and *k_d_* is exemplified in Figure 4. Some results on the surface irregularities, respectively, on the scallop heights on the thread flank *h_max_*, in axial section plane, were discussed in a previous paper [48].

Theoretical models have been tested through simulation on a wide range of variations for the input parameters, as follows: *p* = 2 ÷ 44 mm, *d* = 25 ÷ 200 mm, *z_c_* = 1 ÷ 10, *k_d_* = 1.2 ÷ 1.8, *n_c_* = 300 ÷ 1000 rpm, *n_p_* = 3 ÷ 10 rpm. Also, the simulation results for the output parameters covered a wide range of variation, respectively: *E_pax_* from 2 × 10^−4^ mm up to 0.25 mm, *E_pdm_* from 1 × 10^−4^ mm up to 0.07 mm, *h_max_* from 1 × 10^−3^ μm up to 9 μm, *E_ph0_* from 1 × 10^−4^ mm up to 0.04 mm.

### 2.2. Experimental Conditions

Hypotheses of theoretical modeling, respectively, excluding the influence of factors such as machining dynamics, thermal phenomena, tool edge profile errors, geometrical setting errors, residual errors, material spring back, etc., result in limitations of the theoretical models in accurately predicting the specific machining errors at thread whirling.

Experiments were developed for validating the theoretical models and for establishing empirical mathematical models to describe the influence of the considered input factors on some elements of thread accuracy.

An experimental program was performed under the following research hypotheses:The empirical models follow the trends of machining conditions influences highlighted by the theoretical models;Differences should be registered due to other controllable or uncontrollable factors involved in the process;For optimal adjusting of processing conditions on favorable values of the input parameters, the values of performance parameters, in terms of accuracy and surface roughness, are similar to those obtained by grinding operations, proving the adequacy of thread whirling as a single or final operation for generating medium or large pitch trapezoidal threads;Experimental results, the obtained empirical mathematical models, and the related conclusions have validity for the particular experimented domain, defined through the range of variation of input parameters.

The following micro-geometric accuracy parameters have been experimentally investigated for whirled trapezoidal threads: the thread pitch deviation *E_p_* (μm); the thread flank angle deviation *E_α/2_* (°); the flank profile shape deviation *E_fp_* (mm); and the pitch diameter deviation *E_d2_* (mm). Also, the roughness parameter *R_a_* was investigated, measured in two directions: along the flank profile, in the axial section plane, notated *R_a__*_pr_ and tangent to the thread lead helix, notated *R_a__*_E_.

As analyzed above, the selection of studied influence factors based on the aspects highlighted by the theoretical modeling approach and the current state of the research field. The following input parameters have been considered in experiment planning: the thread external diameter *d* (mm), the thread pitch *p* (mm), the diameter coefficient of whirling process *k_d_*, the rotary speed of whirling head *n_c_* (rpm), and the rotary speed of workpiece *n_p_* (rpm).

To obtain results and information about the effects of all of the input parameters and their interactions, the decision was made to perform 2^5^ full factorial experiments for the first instance. The values selected, on documented criteria, for the levels of the influence factors are presented in Table 1. Considering the difficulty degree in factors’ modifying the structure of the experiment was established as presented in Table 2, where also the corresponding values for machining parameters, cutters’ speed *v* (m/min), and feed rate per cutter *f_z_* (mm/tooth) are shown.

Samples of non-alloy steel for quenching and tempering C45 (SR EN ISO 683-1:2018) were threaded, as shown in Figure 1a, on a normal lathe endowed with four cutters thread whirling equipment. Some sets of cutters and threaded samples are shown in Figure 5.

Measuring conditions for the dimensional accuracy parameters are presented in Figure 6, a Universal Tool Microscope 19 JC (China) being used for allowing the measuring of linear dimensions, in 2 Cartesian coordinates, with 0.001 mm accuracy and of angular dimensions with 30” accuracy. Figure 6a shows the general view of the measuring apparatus. Figure 6b shows how the theoretical axial deviation of the flank profile *E_pax_* (see sub-heading 2.1) is compounded by a theoretical deviation of the flank half angle *E_Tα/2_* and by a theoretical deviation from the rectilinear flank shape, *E_Tfp_*. Theoretical profile exemplified in Figure 6b corresponds to the following values of input parameters: *d* = 36 mm, *p* = 6 mm, *k_d_* = 1.3, *n_c_* = 614 rev/min, *n_p_* = 3.76 rev/min.

Figure 6c illustrates the scheme for thread pitch measurement on thread area corresponding to the stationary thermal regime. The readings *c*1 and *c*2 allowed for the evaluation of the thread pitch deviation *E_p_* through the value of the cumulated pitch deviation *E_3p_*.

Figure 6d show the measurement scheme for *E_d2_* parameter evaluation. The readings *c*1 up to *c8* allowed the evaluation of the pitch diameter deviation *E_d2_*, also considering some residual errors such as the dimensional deviation in workpiece diameter, some geometrical setting errors such as cutting depth setting deviation, and, finally, thermal dilation effects.

In Figure 6e one can see the scheme for measuring parameter *E_α/2_* through *c_α_* readings and the parameter *E_fp_* through *c*1, *c*2, readings. The two errors appear in opposite directions for the left and right flank of the thread, the differences in values being ultimately insignificant.

Roughness measurements were performed using a Taylor-Hobson *Surtronic 3+* instrument, as shown in Figure 7, and measured data have been processed with *Talyprofile* software. For measuring both the investigated roughness parameters, respectively, *R_a__*_pr_ along the flank profile, in axial section plane, and *R_a__*_E_ tangent to the thread lead helix, an adjustable angular positioning system was used. With the necessity of separating the roughness profile from the waviness profile being considered, Gaussian filtering was applied under the *Talyprofile* environment.

Hereby, an example of the roughness measurement result is shown in Figure 8 (i.e., the roughness profile and roughness parameters in Figure 8a, respectively, and the unfiltered profile (yellow line) versus the waviness profile (blue line) in Figure 8b). Results from Figure 8 correspond to roughness parameter *R_a__*_pr_ measured on the threaded sample no. 31 from the experimental plan in Table 2, respectively to the following machining conditions *p* = 10 mm, *d* = 48 mm, *k_d_* = 1,3, *n_c_* = 878 rpm, and *n_p_* = 2.4 rpm.

The results of the 2^5^ experiments and data processing into mathematical matrix models returned information about the effects of all of the input parameters and their interactions, making possible primary process optimization.

Also, a subsequent experimental plan, as in Table 3, was developed on the influences of machining conditions parameters *v* and *f_z_* on thread accuracy in conditions of fixing the other input parameters at the following levels: *p* = 10 mm, *d* = 48 mm, *k_d_* = 1.1. The investigated output parameters were: *E_p_*, *E_d2_*, *R_a__*_pr_, and *R_a__*_E_.

## 3. Results

Experimental data from 2^5^ experiments have been mathematically modeled in matrix form, following the ANOVA method, using customized developed calculus MS Excel sheets. Variance analysis based on Fisher criterion.

The reduced matrix empirical model, containing only the significant effects for the investigated parameters *E_p_*, *E_d2_*, *R_a__*_pr_, and *R_a__*_E_ are exemplified as follows:
(12)$$\begin{array}{l} E_{p}=9,8822+\left[\begin{array}{cc} -2,4072 & 2,4072 \end{array}\right]\cdot Ap+\left[\begin{array}{cc} 1,0097 & -1,0097 \end{array}\right]\cdot Ad+\\ +\left[\begin{array}{cc} 0,5097 & -0,5097 \end{array}\right]\cdot Ak_{d}+\left[\begin{array}{cc} -0,8016 & 0,8016 \end{array}\right]\cdot An_{c}+\left[\begin{array}{cc} -0,9897 & 0,9897 \end{array}\right]\cdot An_{p} \end{array}$$
(13)Ed2=340.4688+[−116.7187116.7187]⋅Ap+[−23.593723.5937]⋅Ad+[21.4062−21.4062]⋅Akd++[−17.343717.3437]⋅Anc+[−26.093726.0937]⋅Anp+Atp⋅[4.2187−4.2187−4.21874.2187]⋅Ad++Atp⋅[2.3437−2.3437−2.34372.3437]⋅Anp+Atd⋅[2.3437−2.3437−2.34372.3437]⋅Anp
(14)Ra_pr=1,5068+[−0,10810,1081]⋅Ap+[−0,05180,0518]⋅Ad++[−0,15180,1518]⋅Akd+[0,4012−0,4012]⋅Anc+[−0,42930,4293]⋅Anp++Atd⋅[0,0281−0,0281−0,02810,0281]⋅Anp+Atkd⋅[−0,03370,03370,0337−0,0337]⋅Anc+Atnc⋅[−0,05250,05250,0525−0,0525]⋅Anp
(15)Ra_E=0,9025+[−0,06250,0625]⋅Ap+[−0,03620,0362]⋅Ad++[−0,08500,0850]⋅Akd+[0,3606−0,3606]⋅Anc+[−0,25620,2562]⋅Anp++Atd⋅[0,0262−0,0262−0,02620,0262]⋅Anp+Atkd⋅[−0,01930,01930,0193−0,0193]⋅Anc++Atkd⋅[0,0250−0,0250−0,02500,0250]⋅Anp+Atnc⋅[−0,05680,05680,0568−0,0568]⋅Anp

Hereby the data sets for the investigated parameters *E_p_*, *E_d2_*, *R_a__*_pr_ and *R_a__*_E_ are exemplified in Table 4.

Graphical representations for the independent factors on *E_d2_* are exemplified in Figure 9. Figure 10 shows the hierarchy of factors’ effects and interactions upon the accuracy parameter *E_d2_*.

Measured data from the experiments presented in Table 3 are shown in Table 5. They were processed with the *DataFit* software tool and mathematical expressions of bi-dimensional dependencies on machining parameters *v* and *f_z_* were obtained. The models for the influences on parameters *E_p_*, *E_d2_*, *R_a__*_pr_, and *R_a__*_E_ are given in Equations (16)–(19).
(16)$$E_{p}=45,8433+14,2989\cdot \mathrm{lg}f_{z}+1,8620\cdot {\mathrm{lg}}^{2}f_{z}-\frac{1769,2930}{v}\cdot (\mu m)$$
(17)$$E_{d_{2}}=98.0393\cdot f_{z}{}^{0,2813}\cdot v^{0,4419}\cdot (\mathrm{\mu m})$$
(18)$$R_{a\_pr}=0,2080+13,1491\cdot f_{z}-0,0114\cdot v-3,7861\cdot 10^{-5}\cdot v^{2}\cdot (\mathrm{\mu m})$$
(19)$$R_{a\_E}=1.3183+7.4574\cdot f_{z}+\frac{533.1008}{v}+\frac{41099.1826}{v^{2}}\cdot (\mathrm{\mu m})$$

Figure 11 shows the response surfaces for the previous exemplified models.

## 4. Discussion

Simulation tests on the theoretical model returned a great number of data and information regarding the influences of the machining conditions, such as the thread pitch *p* (mm), the thread external diameter *d* (mm), the number of cutters on the whirling head *z_c_*, the diameter coefficient of whirling process *k_d_* defined as the ratio between the diameter *D* of cutters’ top edge disposal and *d*, the whirling head tilting angle *β* (°), the rotary speed of whirling head *n_c_* (rpm), and the rotary speed of workpiece *n_p_* (rpm) upon several micro-geometric accuracy elements such as the following: the profile errors on the thread flank and in the axial section plane, symbolized as *E_pax_* (mm), the profile error on the pitch diameter, symbolized as *E_pdm_* (mm); the scallop height on the thread flank and in the axial section plane, symbolized as *h_max_* (μm); and the local error of the helix pitch, symbolized as *E_ph0_* (mm).

Kinematic parameters *n_c_* and *n_p_* proved to have an insignificant effect on theoretical errors *E_pax_* and *E_pdm,_* but strongly influenced *h_max_*. Bigger values for *n_c_* determined smaller values for *h_max_*, the effect of *n_p_* being the opposite. The parameters *n_c_* and *n_p_* did not affect the maximum value of *E_ph0_* but they influenced the waviness heights along the helix thread. Bigger values for *n_c_* determine smaller values for these parameters, the effect of *n_p_* being the opposite.

Dimensional parameters of the thread *p* and *d* have an important influence on *E_pax_* and *E_pdm_*. Bigger values for *d* determine smaller values for *E_pdm_*, the effect of *p* being the opposite. Their influence on *h_max_* is smaller, and in the same direction of increasing *h_max_* for bigger values *p* or *d*.

Diameter coefficient *k_d_* proved an insignificant effect on theoretical errors *E_pax_* and *E_pdm_*, but strongly influenced *h_max_*. Bigger values for *k_d_* determine bigger values for *h_max_*.

Experimental values for the parameter *E_p_* situated in the range of 4 - 19 μm were mainly determined by the geometric errors of the leading mechanism of the machine tool and also containing a component from the thermal phenomena. Parameter *E*_*α*/2_ took values between 2.5′–7′, the importance of an accurate execution and position setting for the cutters. The accuracy parameter *E_fp_* took values in a wider range, respectively, within 2–45 μm. Experimental data show that the empirical models for this parameter followed the theoretically modeled evolutions (abewhile also being affected by process dynamics). The values for parameter *E*_*d*2_ were situated in the range of 120–660 μm, with a maximal reduction of the geometrical errors in the radial position setting for the cutters and the cutting depth setting being highlighted. Parameters of the machining conditions *v* and *f_z_*, but mainly *f_z_*, manifested important influences on parameters *E_p_* and *E*_*d*2_. In contrast, their influence on parameters *E*_*α*/2_ and *E_fp_* was less important.

Roughness parameter *R_a_* measured along the flank profile thein axial section plane, symbolized as *R_a__*_pr_, took values in the range of 0.5–3.2 μm. In contrast, roughness parameter *Ra* measured tangent to the thread lead helix and symbolized as *R_a__*_E_, took values between 0.3 and 2.1 μm, which is an advantage for trapezoidal threads as leading mechanisms.

## 5. Conclusions

Theoretical modeling and simulation of the complex helical surfaces of whirled trapezoidal threads and their specific micro-geometric accuracy related to machining conditions delivered important reliable information useful for predicting real process accuracy and surface quality.

The threaded surface was described as a complex compound surface resulting from intersecting successive ruled helical surfaces corresponding to the cutting edges of the set of cutters from the whirling head.

Numerical simulation results were exemplified, and they highlighted that the theoretical errors of the axial flank profile take maximum values at the bottom and at the top of the thread flank, with minimum values located at a radial position a little smaller than the pitch diameter (since the tilting angle of the whirling head is equal to the lead angle). Tilting angle modification results in a displacement of this radial position of the smallest profile error values.

The dimensional parameters *p* and *d* of the machined threads theoretically influence the flank profile errors and, in less quantity, the scallop height. Process kinematics, considered without the dynamic factors they determine, proved a significant influence on the surface irregularities values and insignificant influence on the flank profile errors.

Practically, the importance of theoretical modeling and simulation is given by the unlimited number of machining conditions that can be simulated, as presented above in Section 2.2, for obtaining valuable referential data in experiment planning or real process behavior predicting. The wide variation ranges for the output parameters’ values resulting from simulation once more highlighted the importance of mastering the influences of process variables on the machining accuracy and optimally adjusting industrial processing conditions for final product quality.

The manner approached in theoretical modeling and in developing the modular, procedural programming of the simulation software tool, starting from the general case and following the way of particular customizing for the certain studied situations, leaves space for further development of the conceived models and future adaption to other particular case studies such as other types and dimension ranges of threads or other helical surfaces and comparative studies with other machining generation variants (such as single point threading, milling, etc.).

Validation experiments were designed and performed and empirical mathematical models were established to highlight the influence of the input factors, such as thread pitch and external diameter, the ratio between the diameter of cutters’ top edge disposal and the thread external diameter, the rotary speed of whirling head, and the rotary speed of workpiece, on some accuracy elements and roughness parameters of the threaded surface.

Regarding the experimental results, important information was obtained from data processing concerning the variation range for the investigated machining accuracy parameters and the influence of the machining conditions upon them.

Experimental values for the accuracy parameters confirmed the research hypotheses and expectations. They allowed us to classify the sample threads into seven to eight classes according to ISO 2903:2016, and the obtained roughness values corresponded to grinding operations, proving the importance of thread whirling as a valuable technology alternative applied as a final or single operation for machining such helical surfaces.

In the future, a more in-depth theoretical approach to developing a more relevant modeling algorithm will concern the authors. This approach will be intended to reduce the simplifying hypotheses and consider other influence factors such as process dynamics, thermal phenomena, material elasticity, etc.

## Figures and Tables

**Figure 1 micromachines-13-01520-f001:**
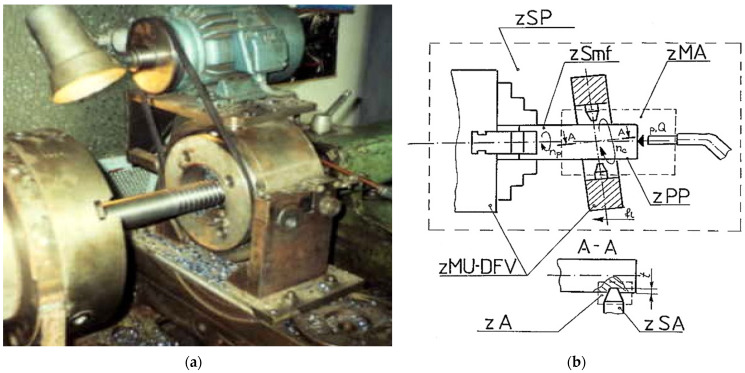
Thread whirling process: (**a**) real process image; (**b**) systemic model for thread whirling process variables classification.

**Figure 2 micromachines-13-01520-f002:**
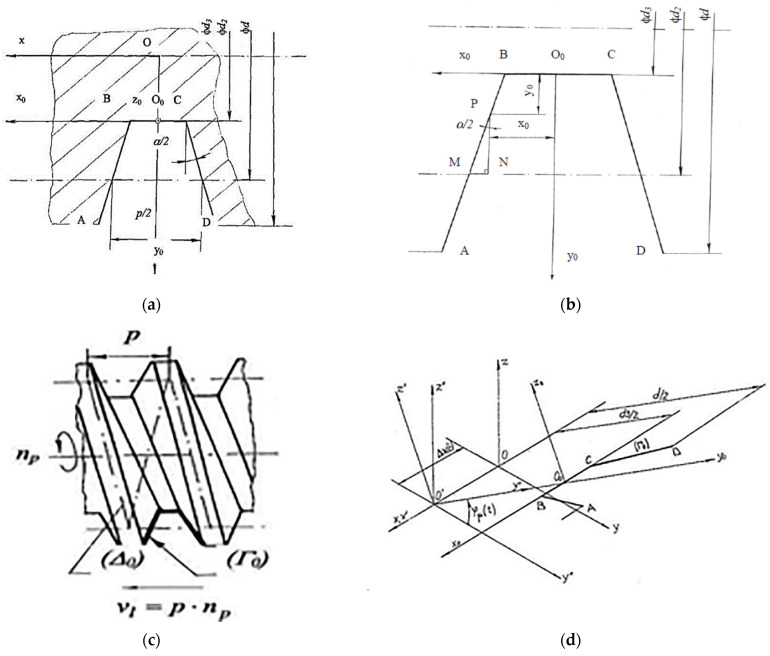
Geometrical schemes for modeling the theoretical surfaces generated by whirling: (**a**) generator profile for trapezoidal threads; (**b**) generator profile representation scheme; (**c**) idealized helical surface (*Σ*_0_) generation; (**d**) coordinate transformation for (*Σ*_0_) modeling; (**e**) coordinate transformation for modeling of theoretical helical surface (*Σ_q_*); (**f**) angular step of cutters’ disposal; (**g**) angular period of theoretical surface generation; (**h**) intersection curve (*Cq*), between theoretical surfaces (*Σ_q_*) and (*Σ*_*q*+1_).

**Figure 3 micromachines-13-01520-f003:**
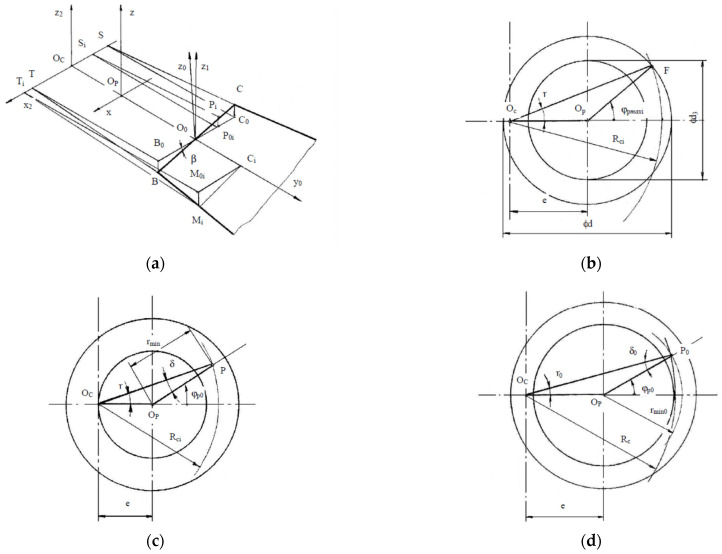
Geometrical schemes for solving particular problems in numerical modeling of the theoretical surfaces generated by whirling: (**a**) calculus of the radius *R_Ci_* of the trajectory of a generator point *M_i_*; (**b**) approximation of contact half-angle *φ_p_*_max*i*_ for *M_i_*; (**c**) approximation of *r_min_* for thread flank; (**d**) approximation of *r_min_* for thread groove bottom; (**e**) calculus of the number of theoretical helical surfaces intersecting a particular axial plane; (**f**) calculus of the number of intersection points between (*Cq*) curves and the axial plane, on the thread flank.

**Figure 4 micromachines-13-01520-f004:**
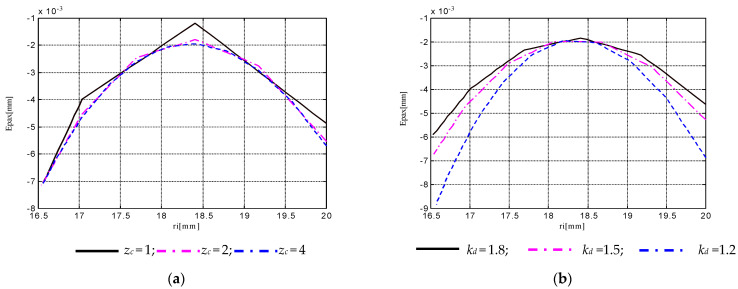
Distribution of the profile errors *E_pax_* along the thread flank profile: (**a**) influence of cutters’ number *z_c_* for *d* = 40 mm; *p* = 6 mm; *k_d_* = 1.4; *β* = 2.95°; *n_c_* = 600 rpm; *n_p_* = 8 rpm; (**b**) influence of diameter coefficient of whirling process *k_d_* for *d* = 40 mm; *p* = 6 mm; *z_c_* = 2; *β* = 2.95°; *n_c_* = 600 rpm; *n_p_* = 8 rpm.

**Figure 5 micromachines-13-01520-f005:**
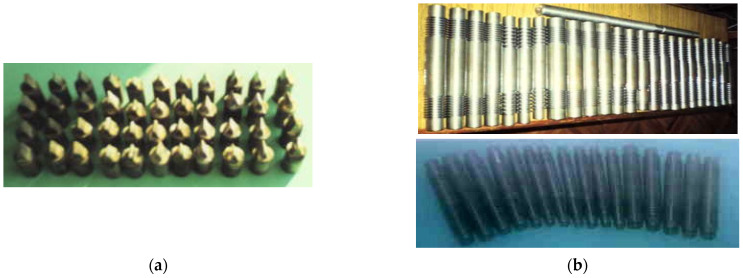
Sets of whirling cutters and threaded samples from the experimental research: (**a**) sets of cutters for trapezoidal threads’ whirling; (**b**) threaded samples.

**Figure 6 micromachines-13-01520-f006:**
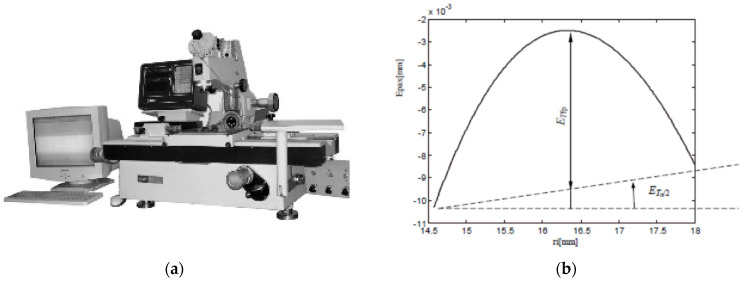
Apparatus and schemes for dimensional accuracy parameters measuring: (**a**) general view on Universal Tool Microscope 19 JC; (**b**) theoretic profile of thread flank, highlighting *E*_*Tα*/2_ and *E_Tfp_*; (**c**) scheme for thread pitch measurement; (**d**) measurement scheme for *E*_*d*2_ parameter evaluation; (**e**) scheme for *E*_*α*/2_ and *E_fp_* measuring.

**Figure 7 micromachines-13-01520-f007:**
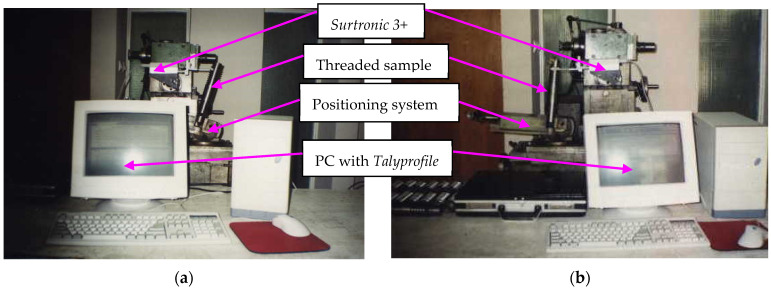
Roughness measuring conditions: (**a**) *R_a__*_pr_ roughness measuring image; (**b**) *R_a__*_E_ roughness measuring image.

**Figure 8 micromachines-13-01520-f008:**
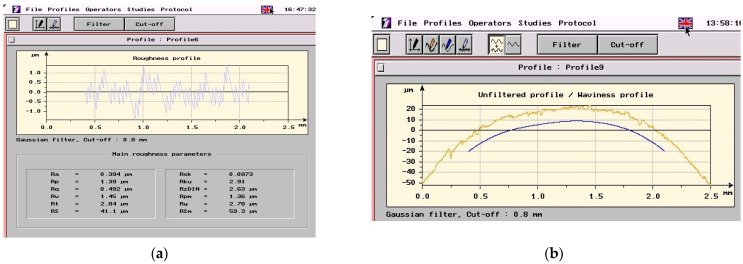
Roughness measurement result example: (**a**) roughness profile and roughness parameters; (**b**) unfiltered profile versus waviness profile.

**Figure 9 micromachines-13-01520-f009:**
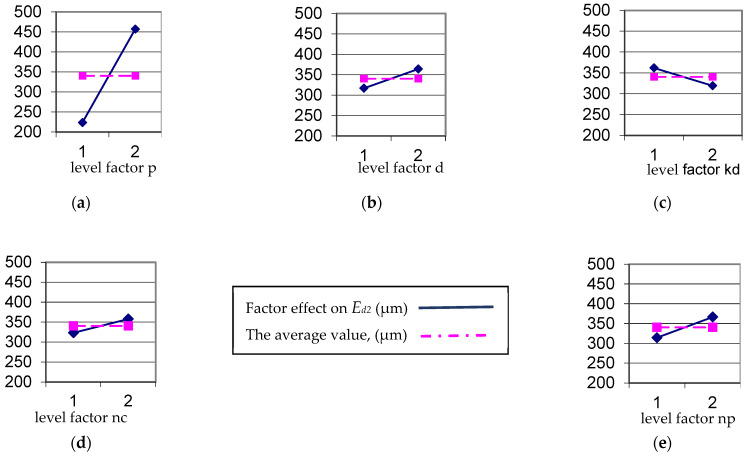
Effects of the independent factors upon parameter *E_d2_*: (**a**) effect of factor *p*; (**b**) effect of factor *d*; (**c**) effect of factor *k_d_*; (**d**) effect of factor *n_c_*; (**e**) effect of factor *n_p_*.

**Figure 10 micromachines-13-01520-f010:**
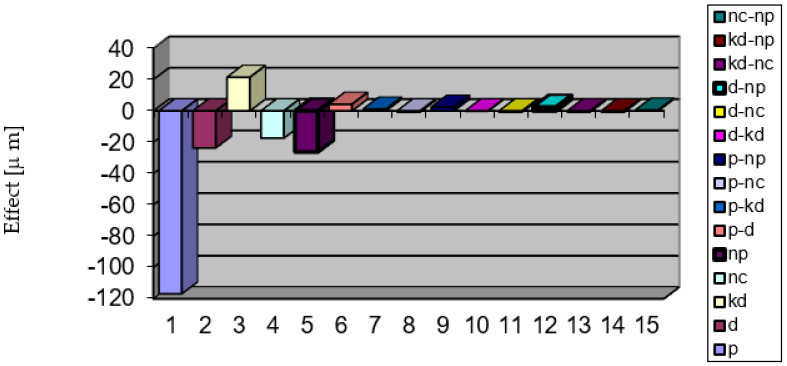
Hierarchy of the effects of factors and of interactions upon parameter *E_d2_*.

**Figure 11 micromachines-13-01520-f011:**
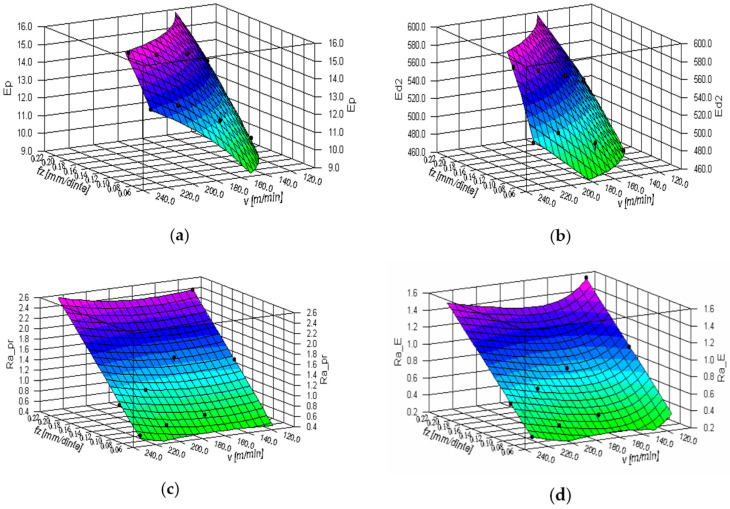
Bi-dimensional dependencies on machining regime parameters *v* and *f_z_* for thread whirling results: (**a**) influence on parameter *E_p_* (μm); (**b**) influence on parameter *E_d2_* (μm); (**c**) influence on parameter *R_a__*_pr_ (μm); (**d**) influence on parameter *R_a__*_E_ (μm).

**Table 1 micromachines-13-01520-t001:** Values for the levels of input factors within the 2^5^ experiment.

Factor Level	Factor				
	*p* [mm]	*d* [mm]	*k_d_*	*n_c_* [rpm]	*n_p_* [rpm]
1	6	36	1.1	614	2.4
2	10	48	1.3	878	3.76

**Table 2 micromachines-13-01520-t002:** The detailed structure of the performed 2^5^ experiment.

No.	Factor Level					*f_x_* [mm/tooth]	*v* [m/min]
	*p*	*d*	*k_d_*	*n_c_*	*n_p_*		
1	1	1	1	1	1	0.11	88.7
2	1	1	1	1	2	0.17	88.7
3	1	1	1	2	1	0.08	117.5
4	1	1	1	2	2	0.12	117.5
5	1	1	2	1	1	0.11	96.1
6	1	1	2	1	2	0.17	96.1
7	1	1	2	2	1	0.08	137.3
8	1	1	2	2	2	0.12	137.3
9	1	2	1	1	1	0.15	107.6
10	1	2	1	1	2	0.23	107.6
11	1	2	1	2	1	0.10	153.9
12	1	2	1	2	2	0.16	153.9
13	1	2	2	1	1	0.15	126.1
14	1	2	2	1	2	0.23	126.1
15	1	2	2	2	1	0.10	180.4
16	1	2	2	2	2	0.16	180.4
17	2	1	1	1	1	0.11	92.1
18	2	1	1	1	2	0.17	92.1
19	2	1	1	2	1	0.08	123.0
20	2	1	1	2	2	0.12	123.0
21	2	1	2	1	1	0.11	99.9
22	2	1	2	1	2	0.17	99.9
23	2	1	2	2	1	0.08	142.9
24	2	1	2	2	2	0.12	142.9
25	2	2	1	1	1	0.15	111.5
26	2	2	1	1	2	0.23	111.5
27	2	2	1	2	1	0.10	159.4
28	2	2	1	2	2	0.16	159.4
29	2	2	2	1	1	0.15	130.0
30	2	2	2	1	2	0.23	130.0
31	2	2	2	2	1	0.10	185.9
32	2	2	2	2	2	0.16	185.9

**Table 3 micromachines-13-01520-t003:** Experiments for investigating influences of machining parameters *v* and *f_z_* on whirling accuracy parameters.

Exp. No.	*n_c_* [rpm]	*n_p_* [rpm]	*v* [m/min]	*f_z_* [mm/tooth]
1	614	2.4	111.5	0.15
2	878	2.4	159.4	0.10
3	1070	2.4	197.6	0.09
4	1241	2.4	229.2	0.07
5	614	3.76	111.5	0.23
6	878	3.76	159.4	0.16
7	1070	3.76	197.6	0.13
8	1241	3.76	229.2	0.11

**Table 4 micromachines-13-01520-t004:** Measured values *E_p_*, *E_d2_*, *R_a__*_pr,_ and *R_a__*_E_., in 2^5^ experiment points.

No.	Factor Level					*E_p_* [μm]	*E_d2_* [μm]	*R_a_pr_* [μm]	*R_a_E_* [μm]
	*p*	*d*	*k_d_*	*n_c_*	*n_p_*				
1	1	1	1	1	1	7.33	185.312	1.08	0.76
2	1	1	1	1	2	8.66	228.750	1.95	1.32
3	1	1	1	2	1	9.33	223.750	0.50	0.,27
4	1	1	1	2	2	10.66	269.062	1.21	0.51
5	1	1	2	1	1	6.66	143.750	1.47	0.94
6	1	1	2	1	2	7.66	184.062	2.32	1.48
7	1	1	2	2	1	8.66	179.062	0.70	0.35
8	1	1	2	2	2	9.66	221.250	1.44	0.74
9	1	2	1	1	1	4.66	221.250	1.15	0.82
10	1	2	1	1	2	7.00	274.062	2.19	1.45
11	1	2	1	2	1	6.33	256.562	0.52	0.25
12	1	2	1	2	2	9.00	311.250	1.35	0.63
13	1	2	2	1	1	4.00	179.062	1.51	1.03
14	1	2	2	1	2	6.33	228.750	2.60	1.74
15	1	2	2	2	1	5.33	211.250	0.82	0.34
16	1	2	2	2	2	8.33	262.812	1.57	0.81
17	2	1	1	1	1	11.00	405.000	1.40	0.93
18	2	1	1	1	2	14.33	457.812	2.21	1.47
19	2	1	1	2	1	13.66	440.312	0.68	0.35
20	2	1	1	2	2	15.33	495.000	1.48	0.68
21	2	1	2	1	1	11.00	367.812	1.65	1.09
22	2	1	2	1	2	13.00	417.500	2.64	1.74
23	2	1	2	2	1	13.00	400.000	0.95	0.40
24	2	1	2	2	2	14.33	451.562	1.60	0.83
25	2	2	1	1	1	9.66	457.812	1.38	0.95
26	2	2	1	1	2	13.00	520.000	2.42	1.57
27	2	2	1	2	1	11.66	490.000	0.64	0.36
28	2	2	1	2	2	14.66	554.062	1.52	0.76
29	2	2	2	1	1	9.00	420.000	1.77	1.08
30	2	2	2	1	2	12.00	479.062	2.79	1.84
31	2	2	2	2	1	11.00	449.062	1.02	0.42
32	2	2	2	2	2	10.00	510.000	1.69	0.97

**Table 5 micromachines-13-01520-t005:** Measured values *E_p_*, *E_d2_*, *R_a__*_pr_, and *R_a__*_E_ for investigating influences of machining parameters *v* and *f_z_* on whirling accuracy.

Exp. No.	*v* [m/min]	*f_z_* [mm/tooth]	*E_p_* [μm]	*E_d2_* [μm]	*R_a_pr_* [μm]	*R_a_E_* [μm]
1	111.5	0.15	9.66	460	1.38	0.95
2	159.4	0.10	11.66	490	0.64	0.36
3	197.6	0.09	13.00	510	0.59	0.33
4	229.2	0.07	13.33	510	0.56	0.30
5	111.5	0.23	13.00	520	2.42	1.57
6	159.4	0.16	14.66	550	1.52	0.76
7	197.6	0.13	15.33	570	1.12	0.66
8	229.2	0.11	16.00	585	0.99	0.59

## Data Availability

Data are available on request from the authors.

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
