# Peer review of "Influence of Machining Conditions on Micro-Geometric Accuracy Elements of Complex Helical Surfaces Generated by Thread Whirling"

_micromachines, 2022, doi:10.3390/mi13091520_

Round 1

Reviewer 1 Report

This work study the complex helical surface generated by thread whirling through theoretical model, numerical simulation and experiment. It provides a comprehensive analysis to help understanding and controlling the complex machine process. I have no questions.

Reviewer 2 Report

Dear Authors,

The manuscript is poorly presented. Although it has vital information, the authors fail to define the current problem, and evidence to prove the accuracy of the proposed model.

Here are a few comments for improvement

1. L 14, remove 'the'

2. Introduction - please take off the figures. It looks like a thesis introduction. The introduction should set the background (literature). If there are any research inputs to set the story (including any extra inputs to the figures), should be under a separate heading. 

3. Headings 1 & 2 should be modified. 

4. Heading 2.1 - again sounds like setting up the need of this research, which should be under Introduction

5. The hypotheses - L304 and L305 sound general. What made such a broad hypothesis? Any references?

6. Please leave a space after L307 for better understanding.

7. Why choose trapezoidal thread? 

8. L640 - how can you conclude this is a more precise model? have you done any experimental comparison study with the existing mathematical models?

89.  L642 -643 re-write - Contraditing with the previous sentences.

Round 2

Reviewer 2 Report

The authors have made necessary corrections